# Dosing trajectories of antihypertensive agents among preterm neonates: A retrospective, cross-sectional analysis

Mary-Carty Pittman [1¤a], Alejandro D. Perez [1¤b], Kaniz Afroz Tanni[2], Keia R. Sanderson[3], Jieun Park [4], Daniel I. Feig[5], Matthew M. Laughon[6], Matthew Shane Loop [2,7]*

**1** UNC Eshelman School of Pharmacy, University of North Carolina at Chapel Hill, North Carolina, United States of America, **2** Department of Health Outcomes Research and Policy, Harrison College of Pharmacy, Auburn University, Alabama, United States of America, **3** Division of Nephrology and Hypertension, University of North Carolina School of Medicine, University of North Carolina at Chapel Hill, North Carolina, United States of America, **4** Harrison College of Pharmacy, Auburn University, Alabama, United States of America, **5** Division of Pediatric Nephrology, Heersink School of Medicine, University of Alabama at Birmingham, Alabama, United States of America, **6** Division of Neonatal-Perinatal Medicine, University of North Carolina School of Medicine, University of North Carolina at Chapel Hill, North Carolina, United States of America, **7** Division of Pharmacotherapy and Experimental Therapeutics, UNC Eshelman School of Pharmacy, University of North Carolina at Chapel Hill, North Carolina, United States of America

¤a Current work address: WakeMed Health & Hospitals, New Bern Avenue, Raleigh, North Carolina.
¤b Current work address: University of North Carolina Medical Center, 101 Manning Drive, Chapel Hill, North Carolina.
* matthew_loop@unc.edu

## Abstract

### Introduction

Preterm infants are at increased risk for hypertension due to incomplete organ development. With no established guidelines for treating hypertension in this population, clinicians rely on their experience, and little is known regarding those treatment decisions. This study described dosing trajectories for the three most common antihypertensive agents in preterm infants.

### Methods

This retrospective, cross-sectional study examined preterm neonates (gestational age at birth < 37 weeks, postmenstrual age < 44 weeks) treated with antihypertensive medications at the University of Alabama at Birmingham (UAB) Medical Center. The study identified the three most common antihypertensive agents and identified common dosing patterns over time using the functional K-means algorithm. We then compared demographic and clinical information across these clusters.

### Results

The study included 87 patients across 93 visits. The three most common antihypertensive agents were propranolol (61%), captopril (8.8%), and esmolol (12%). Median

**Data availability statement:** The study authors are prevented from sharing the data publicly due to restrictions placed upon use of the data by the data owners. To initiate a request for the dataset, please contact ResearchData@uabmc.edu and reference request "i2b2.lds.inpatient infants receiving antihypertensives w/dx codes <37 pt set". The corresponding author (matthew_loop@auburn.edu) will facilitate obtainment of the data for reasonable requests. To obtain access to the data, you can also contact the UAB CCTS Clinical Research Informatics Team (mwyatt@uabmc.edu).

**Funding:** We would like to acknowledge the University of Alabama at Birmingham Medical Center for providing the retrospective data used in this study. Research reported in this publication was supported by the National Center for Advancing Translational Sciences of the National Institutes of Health under award number UL1TR003096. The content is solely the responsibility of the authors and does not necessarily represent the official views of the National Institutes of Health. The funders had no role in study design, data collection and analysis, decision to publish, or preparation of the manuscript.

**Competing interests:** The authors have declared that no competing interests exist.

treatment durations were 292 hours for propranolol, 186 hours for captopril, and 68 hours for esmolol. Propranolol was often initiated at a dose that was maintained with few dosing changes. Patients needing higher doses of propranolol were generally younger gestational age and spent longer in the hospital. Captopril doses started low and increased over time, likely due to safety concerns. Esmolol showed the most variable dosing trajectories.

## Conclusions

Propranolol is often initiated at a target dose and maintained with less titration, while both captopril and esmolol are titrated more often. Younger gestational age patients typically required higher doses of propranolol.

---

## Introduction

Hypertension affects 1–2% of infants in the Neonatal Intensive Care Unit (NICU) [1]. Preterm infants have several unique risk factors that place them at higher risk for developing hypertension, including renal artery thrombosis from umbilical catheterization, medication-related complications, parenteral nutrition (volume, calcium, and salt excess), extracorporeal membrane oxygenation (ECMO), bronchopulmonary dysplasia, and patent ductus arteriosus [2]. When hypertension is present, it presents via nonspecific symptoms such as irritability, feeding difficulties, and vomiting, making hypertension problematic to identify and diagnose [3]. However, it is defined by the American Academy of Pediatrics as a systolic blood pressure (SBP) greater than the 95th percentile based on gestational age, birthweight, and sex on three different occasions [4,5]. The complications of untreated hypertension in preterm infants can be wide-ranging compared to those of adults due to incomplete organ development [2] and could include retinopathy [6], cardiomyopathy [7], and mortality [8]. A prospective study of 34 neonates treated for systemic hypertension found that 29% had target organ damage, most commonly involving the cardiovascular system [9]. Therefore, to prevent such complications once hypertension is identified, appropriate treatment in preterm infants is critical.

Treating hypertension with a specific pattern of dose administration over time (i.e., a "dose trajectory") in preterm infants is challenging because there are no treatment guidelines or formal statements from the American Academy of Pediatrics, American College of Cardiology, or American Heart Association. References such as Lexicomp and NeoFax also do not have specific dosing recommendations for these agents for preterm infants [10–15]. Guidelines do not exist because randomized controlled trials are rarely conducted in this population. Of randomized controlled trials in children submitted to the Food and Drug Administration via the Pediatric Exclusivity Program from 1998 to 2005, only 0.01% of participants were preterm infants [16]. Organizations such as the Pediatric Trials Network and the Neonatal Research Network are making efforts to conduct more randomized controlled trials in this population, but the studies to date do not include dosing strategies over time [17]. Information specific

to appropriate dosing of agents is limited, and the currently accepted treatment strategy for hypertension is to use different doses for mild, moderate, and severe hypertension in preterm infants [4,18–20]. Relying on clinical expertise to dose agents for the treatment of a complicated condition with potentially severe complications leads to a lack of standardization and different outcomes for patients.

Therefore, we conducted a retrospective, cross-sectional study that addressed this gap in the literature by describing the dosing strategies over time (dosing trajectories) of antihypertensive agents used in preterm neonates (< 37 weeks gestation and < 44 weeks postmenstrual age). We analyzed electronic medical record (EMR) data from the University of Alabama at Birmingham (UAB) Medical Center to identify clusters of patients based on antihypertensive dosing trajectories. Identifying the treatment patterns for different groups of patients is important because treatment strategies may vary based on measurable patient characteristics.

## Methods

### Target population & sampling strategy

Our ideal target population was preterm (born < 37 weeks gestation) neonates (< 44 weeks postmenstrual age) treated for hypertension. Postmenstrual age is the age of the infant since the mother's last menstrual period, and it includes both time spent in the womb (gestational age) plus time outside the womb (i.e., postnatal age). We chose this population because preterm infants have higher risk for hypertension [21,22], and hypertension often resolves as infant age increases [23]. We used receipt of an antihypertensive medication as a proxy for receipt of treatment for hypertension. A previous study of a patient population born <= 32 weeks gestational age and <= 1500 g with no congenital anomalies found that receipt of an antihypertensive medication had a positive predictive value of 75% for having a mention of hypertension in the medical record [24]. We sampled from this population by performing a retrospective cross-sectional study of EMR data from the UAB Medical Center. We identified participants between early 2010 and December 31, 2022, who: 1) were admitted as inpatients at UAB Medical Center; 2) were born prior to 37 weeks gestation; 3) were neonates during their visit (postmenstrual age at discharge < 44 weeks); and 4) received any of the following antihypertensive medications intravenously or orally: vasodilators (hydralazine, nitroprusside, clonidine), angiotensin-converting enzyme (ACE) inhibitors (enalapril, captopril), beta-blockers (propranolol, labetalol, esmolol, atenolol, metoprolol), and calcium channel blockers (amlodipine, isradipine, nifedipine, nicardipine). We did not include diuretics in order to be consistent with prior literature on antihypertensive drug exposure [24], since including diuretics would have likely identified many cases of other conditions such as bronchopulmonary dysplasia. Patients were excluded if they received intravenous (IV) propranolol. Patients were excluded if they received IV propranolol because oral propranolol, the dominant route, has a bioavailability of approximately 25%. Therefore, oral and IV dose trajectories do not result in equivalent exposure to propranolol and could not be treated as equivalent in the clustering analysis. Only two patients received IV propranolol. Patients were also excluded if they had a discharge diagnosis of hemangioma in any position, as hemangioma is another indication for the use of propranolol in this population.

### Outcome

The primary outcome of this study was the weight-based dosing trajectory of the antihypertensive agent administered. The doses were collected at every administration during the inpatient visit on a minute-level basis. The doses for each agent were standardized to mg/kg for oral medications and to mg/kg/minute for intravenous (IV) medications. Weight was assigned as the weight measured closest in time to the administration of the antihypertensive agent. If there was a tie between two or more weight measurements in time since administration, then the weights were averaged. To describe the dosing trajectories, the doses of oral medications were set to be constant for each hour until an additional dose was given or a dose change was made. After the last dose received by a given patient during follow up was administered, we kept

that dose constant until six hours after the final administration of the drug, at which point the dose was set to 0 mg/kg. An interval of six hours was chosen because this was the most frequent dosing interval, especially for propranolol which was the most prevalent medication.

## Other covariates

We also collected information on sex, race (Asian, Black or African American, Hispanic or Latino, Other, or White), ethnicity (Hispanic/Latino or Non-Hispanic/Latino), gestational age at birth (weeks), birthweight (grams), postnatal age at discharge (days), neonatal hypertension (discharge code of ICD9 401.9 or ICD10 I10 or P29.2), bronchopulmonary dysplasia (ICD9 770.7 or ICD10 P27.1), acute kidney injury (ICD9 584.9 or ICD10 N17.1), congenital heart disease (ICD9 745.0, 745.10, 745.11, 745.2, 745.3, 745.4, 745.5, 745.60, 745.61, 745.69, 746.01, 746.1, 746.2, 746.3, 746.7, 747.10, 747.11, 747.41 or ICD10 Q20.0, Q20.1, Q20.3, Q20.4, Q21.0, Q21.1, Q21.2, Q21.3, Q22.0, Q22.4, Q22.5, Q23.0, Q23.4, Q25.1, Q25.2, Q25.21, Q25.4, Q26.2), and pulmonary hypertension (ICD9 747.83 or ICD10 P29.30). To describe the dosing trajectories, we also included postnatal age at first dose, minimum dose, maximum dose, duration of treatment, mean dose increase, and mean time to dose increase.

## Statistical methods

Descriptive statistics were summarized for the patients. Continuous variables were summarized using the median (quartile 1, quartile 3), and categorical variables were summarized using the frequency with percentage of each level. The prevalence of each antihypertensive agent was estimated across all patient encounters included in the study. For the three most prevalent agents, the longitudinal dose changes were graphed using step-function plots. The selection of the three most prevalent agents was somewhat arbitrary but also based on review of the prevalences of the agents and identifying a sharp drop off in prevalence after the third agent. Doses for each agent were plotted versus time since the first administration of that agent with "steps" signaling a change in dose. Clusters of patients were then identified based on these step-function plots.

To perform the clustering of these dosing trajectories, we used the functional K-means algorithm, implemented in the kml package in R [25–27]. We gave careful consideration to choosing what length of follow-up time to use for the analysis. The K-means algorithm required that all patients have the same treatment duration, but some patients were treated longer than others. For patients with shorter treatment durations, choosing a longer follow-up time would create many values of "0 mg/kg" for these patients. We did not want to include a follow up time that was so long that few participants were still treated at the end, thus assigning a value of 0 mg/kg or 0 mg/kg/min for many values of the curve for a large number of participants. This would have put participants in the same cluster whose dose trajectories looked very different in the early part of treatment simply because they were treated for a shorter amount of time. Therefore, we used the following approach. First, we considered a span of different possible windows. At each possible window, we then computed the pairwise distances between each curve. Second, we calculated the standard deviation of those distances. Finally, we chose the window that maximized the standard deviation of those distances for each drug. For esmolol, there appeared to be an unrealistically short time frame at which maximum variance was reached (1 hour), so we chose an approximate local maximum at 50 hours to increase the number of observations for esmolol dose trajectories. The cutoffs for the time windows were 868.3 hours for propranolol, 336.83 hours for captopril, and 50 hours for esmolol.

The analysis was performed assuming 2, 3, and 4 clusters for each agent. The "max-dist" algorithm was used to assign initial centroids within the kml package. This algorithm is more robust to uneven cluster sizes than the default algorithm of randomly assigning the centroids. It initially chooses centroids that are separated by the maximum distance possible, allowing for more accurate assignment of patients to clusters [28,29] After the assignment of each dose trajectory to a cluster, descriptive statistics on key demographic and clinical features were calculated, including gender, race, ethnicity, gestational age at birth (weeks), birthweight (g), and age at discharge (days). Null hypothesis significance tests for

differences in these variables across cluster assigned were performed using Pearson's $\chi^2$ test, Fisher's exact text, or the Kruskal-Wallis rank sum test of medians as appropriate.

This study was determined to be Not Human Subjects Research by Auburn University and the University of Alabama at Birmingham. The data from these electronic medical records was delivered from the University of Alabama at Birmingham to Auburn University on January 4th, 2023. Study investigators never had access to information that could identify participants, either during or after data collection. The study authors are prevented from sharing the data publicly due to restrictions placed upon use of the data by the data owners. To initiate a request for the dataset, please contact Research-Data@uabmc.edu and reference request "i2b2.lds.inpatient infants receiving antihypertensives w/dx codes <37 pt set". The corresponding author (matthew_loop@auburn.edu) will facilitate obtainment of the data for reasonable requests.

## Results

Of the initial 751 patient visits for 146 unique patients we received from UAB, our study included 93 patient visits for 87 unique patients who met our inclusion criteria. Figure S1 in S1 File shows a flow diagram of patient exclusions. Most of the exclusions of records from UAB were because the records did not fit the target population of interest for the study (i.e., data cleaning). The majority of patients were male (60%) with a median (1st quartile, 3rd quartile) gestational age of 32.1 (28.5, 35.0) weeks, birthweight of 1620 (890, 2310) g, age at discharge from first encounter of 40 (19, 88) days of life (Table 1). Other comorbidities present in these patients included bronchopulmonary dysplasia (32.0%), patent ductus arteriosus (35.0%), pulmonary hypertension (9.8%), and acute kidney injury (3.7%). Neonatal hypertension was formally diagnosed in only 11% of patients. A large proportion of participants were missing birthweight (55%).

The three most prevalent antihypertensive agents used in preterm infants across visits were propranolol (61%), captopril (8.8%), and esmolol (12%) (see Table S1 in S1 File). Twelve patients received both propranolol and esmolol, with esmolol always preceding propranolol. Two patients received both propranolol and captopril within the same visit, with captopril preceding propranolol on both occasions. Table S2 in S1 File shows the same demographics analysis as Table 1, but by presence of a diagnosis for neonatal hypertension. Assuming a false positive rate of 0.05, the only differences we could detect statistically were a higher percentage of Hispanic/Latino patients among those with a hypertension diagnosis (25%) versus without a diagnosis (0%), as well as a higher prevalence of discharge diagnosis codes for bronchopulmonary dysplasia among patients with a diagnosis of neonatal hypertension (67%) versus no diagnosis (27%). Table 2 shows the summary statistics on dosing and dosing intervals for these three drugs. Mean time to dose increase and mean dose increase are missing for treatment regimens where the mass load (mg) of the drug was never changed.

Seventy treatment courses of propranolol, 10 of captopril, and 14 of esmolol were included in the clustering analysis. The three agents exhibited distinct dosing patterns. Although we will show the clustering results for 2–4 clusters in the figures, we discuss only the results from the 3-cluster analysis in the text and in summary tables for brevity. Figs 1–3 depict the dosing trajectories for each agent prior to clustering and after dividing into 2, 3, and 4 clusters with different colors representing the different cluster assignments.

For propranolol, patients in cluster A (Fig 1, 3-cluster graph) were often initiated on a dose of 0.5 to 1.0 mg/kg, while cluster B had a higher prevalence of initial doses $\leq$ 0.5 mg/kg. Cluster C tended to initiate at 1.0 to 1.5 mg/kg. Few dose titrations were seen. Rather, patients were started at the desired dose, which was roughly maintained for the duration of treatment. Slow decreases in dose tended to reflect the weight of the infant changing, as opposed to the mass load (mg) of the drug delivered changing. Median gestational age at birth was statistically significantly different across clusters via a Kruskal-Wallis rank sum test (p = 0.029), with cluster C patients having younger gestational age at birth (29.5 weeks) than cluster A (34.0 weeks) or cluster B (33.6 weeks) (Table 3). Cluster C patients also were discharged from the hospital at older median postnatal ages (53 days) than cluster A (32 days) or cluster B (22 days) (p = 0.013). No other variables were statistically significantly different across the clusters, but birthweight appeared to meaningfully vary across clusters, with

**Table 1. Baseline demographic and clinical characteristics for all patients.**

| Characteristic | N = 87[1] |
|---|---|
| **Gender** | |
| Female | 34 (40) |
| Male | 52 (60) |
| Missing | 1 |
| **Race** | |
| Asian | 1 (1.2) |
| Black or African American | 32 (40) |
| Hispanic or Latino | 0 (0) |
| Other | 1 (1.2) |
| White | 47 (58) |
| Missing | 6 |
| **Ethnicity** | |
| Hispanic/Latino | 2 (2.5) |
| Non-Hispanic/Latino | 78 (98) |
| Missing | 7 |
| **Gestational age at birth (weeks)** | 32.1 (28.5, 35.0) |
| **Birthweight (g)** | 1620 (890, 2310) |
| Missing | 46 |
| **Age at discharge (days)** | 40 (19, 88) |
| **Neonatal Hypertension** | 9 (11) |
| Missing | 5 |
| **Bronchopulmonary Dysplasia** | 26 (32) |
| Missing | 5 |
| **Patent Ductus Arteriosus** | 29 (35) |
| Missing | 5 |
| **Acute Kidney Injury** | 3 (3.7) |
| Missing | 5 |
| **Congenital heart disease** | 44 (54) |
| Missing | 5 |
| **Pulmonary hypertension** | 8 (9.8) |
| Missing | 5 |

[1]n (%); Median (Q1, Q3)

the lowest weight babies born in cluster C. Patients in cluster C also had longer hospital length of stays (53 days old at discharge versus 32 days in Cluster A or 22 days in Cluster B), which correlated with their longer duration of treatment.

In the 3-cluster graphs for captopril (Fig 2), doses in Cluster A (n = 5) were generally ≤ 0.2 mg/kg and showed rapid cessation of treatment. Cluster B patients (n = 3) were given initial doses of 0.1 to 0.2 mg/kg and increased but never reached the doses reached by Cluster C patients (n = 2). All the null hypothesis significance tests of differences among the clusters were inconclusive (Table 4).

For esmolol, the Cluster A patients (n = 9) in the 3-cluster assignments tended to receive either: (1) shorter periods of treatment duration; or (2) lower doses than participants in Clusters B (n = 3) or C (n = 2). Cluster B patients appeared to be treated for longer periods of time than Cluster C patients. Cluster C patients tended to start at higher doses (> 0.3 mg/kg/

**Table 2. Summary statistics of dosing regimens for captopril, esmolol, and propranolol.**

| Characteristic | Propranolol N=70[1] | Captopril N=10[1] | Esmolol N=14[1] |
|---|---|---|---|
| **Postnatal age at first administration of this antihypertensive drug (days)** | 15 (5, 38) | 64 (12, 90) | 2 (2, 19) |
| **Maximum dose during treatment[2]** | 2.72 (2.07, 3.92) | 0.60 (0.21, 1.98) | 0.34 (0.20, 0.41) |
| **Minimum dose during treatment[2]** | 1.14 (0.74, 1.83) | 0.17 (0.10, 0.33) | 0.05 (0.00, 0.06) |
| **Duration of treatment (hours)** | 292 (89, 812) | 186 (32, 358) | 68 (34, 127) |
| **Mean time to dose in mg increase (hours)** | 172 (58, 342) | 47 (37, 104) | 10 (7, 17) |
| Missing | 32 | 4 | 2 |
| **Mean dose increase (mg)** | 0.7 (0.4, 1.0) | 0.4 (0.1, 0.5) | 11.1 (4.7, 14.5) |
| Missing | 35 | 4 | 3 |

[1]Median (Q1, Q3); [2]mg/kg/day for captopril and propranolol, mg/kg/min for esmolol.

Propranolol dose (mg/kg) over time

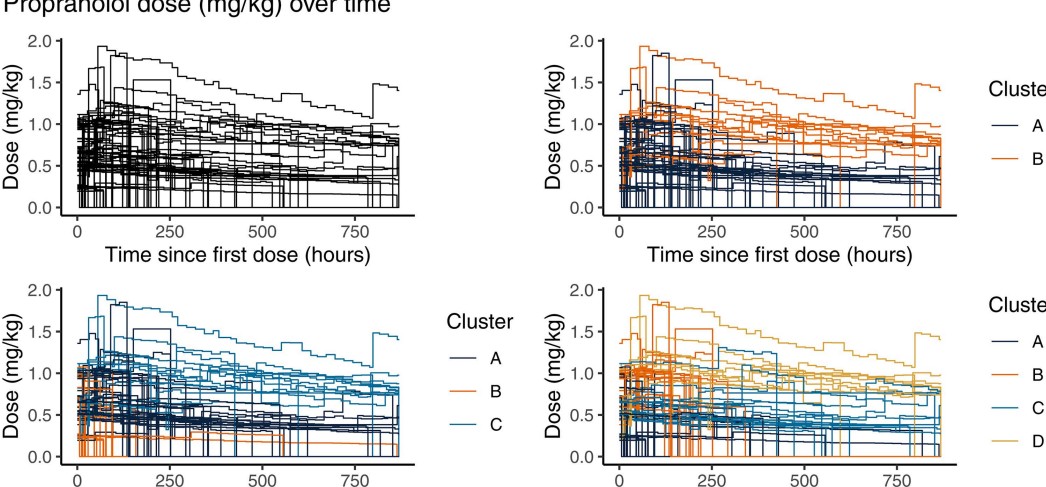

**Fig 1. Step-function plots of longitudinal dose changes for propranolol separated by cluster assignment analyzed using 2, 3, and 4 clusters.**

min) and were titrated more rapidly than cluster B patients (Fig 3). All the null hypothesis significance tests for differences among the clusters were inconclusive Table 5.

While we have included null hypothesis significance tests for among cluster comparisons, many of which were not statistically significant, we remind the reader that the appropriate interpretation of a non-significant hypothesis test is that the differences were "inconclusive." To conclude that there were "no differences" among clusters, we would have had to conduct a formal hypothesis test of equivalence. The small sample size of some of our clusters led to null hypothesis significance tests that could neither confirm nor disconfirm group differences.

## Discussion

Our study found that propranolol, captopril, and esmolol were the most used antihypertensive agents in preterm infants at a large, academic medical center. K-means clustering was performed to identify groups of patients based on their dosing trajectories. The analysis revealed different dosing trajectories for each agent. Propranolol was generally initiated at the desired dose, while captopril was often started at a low dose and titrated up over time. Esmolol had variable starting

Captopril dose (mg/kg) over time

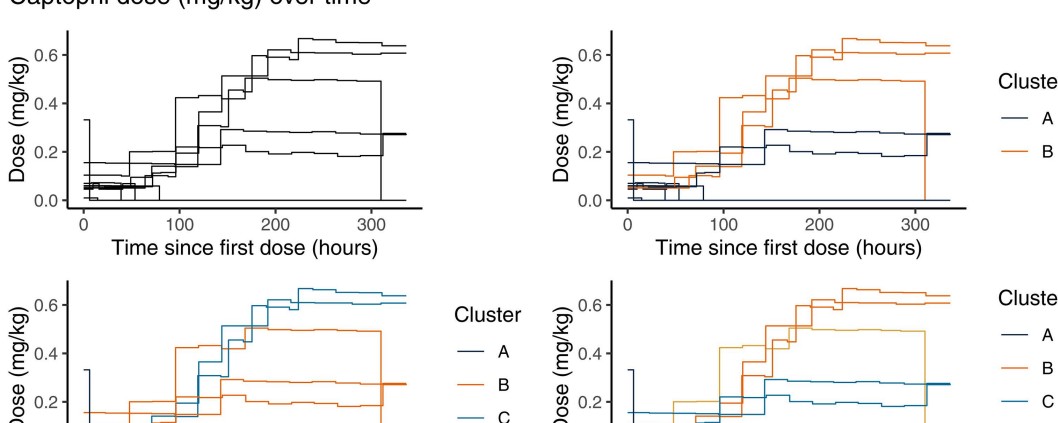

**Fig 2. Step-function plots of longitudinal dose changes for captopril separated by cluster assignment analyzed using 2, 3, and 4 clusters.**

Esmolol dose (mg/kg/min) over time

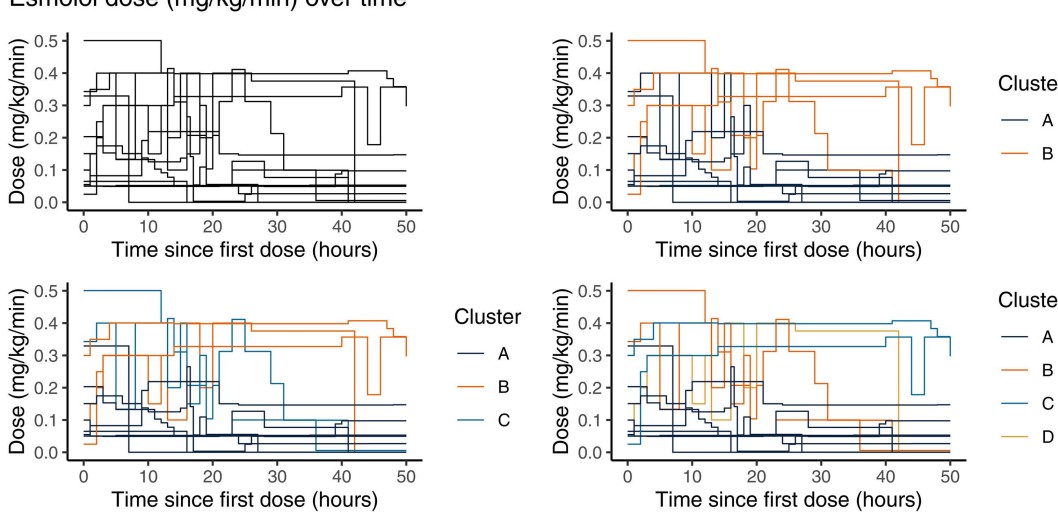

**Fig 3. Step-function plots of longitudinal dose changes for esmolol separated by cluster assignment analyzed using 2, 3, and 4 clusters.**

doses and large dose changes. The highest doses of propranolol were used for infants born at younger gestational ages who required longer hospital stays before discharge.

Previous literature cites hydralazine, captopril, and enalapril as the most common antihypertensive agents used to treat neonatal hypertension.[12] However, we found the most prevalent agents were propranolol, captopril, and esmolol in this single-center study. This difference in prevalence may be due to institutional preferences or availability. The results did, however, coincide with the current practice of using esmolol, an intravenous agent, acutely, and captopril or propranolol for the maintenance of blood pressure [18].

**Table 3. Summary statistics across patient clusters for dosing patterns of propranolol. Results are presented for the 3-cluster models.**

| Characteristic | A N=36[1] | B N=21[1] | C N=13[1] | p-value[2] |
|---|---|---|---|---|
| **Gender** | | | | 0.2 |
| Female | 12 (34) | 12 (57) | 7 (54) | |
| Male | 23 (66) | 9 (43) | 6 (46) | |
| Missing | 1 | 0 | 0 | |
| **Race** | | | | 0.070 |
| Asian | 0 (0) | 0 (0) | 1 (8.3) | |
| Black or African American | 15 (45) | 5 (25) | 7 (58) | |
| Hispanic or Latino | 0 (0) | 0 (0) | 0 (0) | |
| Other | 0 (0) | 1 (5.0) | 0 (0) | |
| White | 18 (55) | 14 (70) | 4 (33) | |
| Missing | 3 | 1 | 1 | |
| **Ethnicity** | | | | >0.9 |
| Hispanic/Latino | 0 (0) | 0 (0) | 0 (0) | |
| Non-Hispanic/Latino | 35 (100) | 19 (100) | 12 (100) | |
| Missing | 1 | 2 | 1 | |
| **Gestational age at birth (weeks)** | 34.0 (31.2, 35.7) | 33.6 (32.0, 35.0) | 29.5 (29.0, 31.5) | 0.029 |
| **Birthweight (g)** | 2243 (1350, 3200) | 1883 (1658, 2500) | 1600 (890, 3200) | 0.6 |
| Missing | 18 | 9 | 6 | |
| **Age at discharge (days)** | 32 (15, 66) | 22 (11, 60) | 53 (41, 107) | 0.013 |
| **Neonatal Hypertension** | 0 (0) | 1 (5.0) | 0 (0) | 0.5 |
| Missing | 1 | 1 | 0 | |
| **Bronchopulmonary Dysplasia** | 7 (20) | 5 (25) | 3 (23) | >0.9 |
| Missing | 1 | 1 | 0 | |
| **Patent Ductus Arteriosus** | 10 (29) | 6 (30) | 6 (46) | 0.5 |
| Missing | 1 | 1 | 0 | |
| **Congenital heart disease** | 17 (49) | 10 (50) | 9 (69) | 0.4 |
| Missing | 1 | 1 | 0 | |
| **Pulmonary hypertension** | 2 (5.7) | 1 (5.0) | 2 (15) | 0.5 |
| Missing | 1 | 1 | 0 | |

[1] n (%); Median (Q1, Q3).

[2] Pearson's Chi-squared test; Fisher's exact test; Kruskal-Wallis rank sum test.

Propranolol was the most prevalent agent and was initiated at its desired dose rather than starting low and increasing over time, possibly due to physicians' confidence in its safety and efficacy profile. In addition, higher weight-based doses were used for longer durations in patients born more preterm with lower birth weights and more severe hypertension. However, its use is generally limited to those who can tolerate oral medication. The recommended starting dose in neonates is 0.25 to 0.5 mg/kg every 8 hours, which corresponds to the strategy implemented in Cluster A [11]. Clusters B and C were initiated on doses higher than recommended but no patient received a dose higher than the recommended maximum dose of 5 mg/kg/day.

Captopril's uniformity in low starting dose and subsequent stepwise increases over time can perhaps be attributed to concerns about using ACE inhibitors in preterm infants due to potential adverse effects such as hypotension, acute renal failure, neurological complications, and potential effects on renal maturation [20,30,31]. These concerns can also explain why captopril was reserved for older patients whose renal systems were more developed. Some patients were started on

**Table 4. Summary statistics across patient clusters for dosing patterns of captopril. Results are presented for the 3-cluster models.**

| Characteristic | A N = 5[1] | B N = 3[1] | C N = 2[1] | p-value[2] |
|---|---|---|---|---|
| **Gender** | | | | 0.2 |
| Female | 3 (60) | 0 (0) | 0 (0) | |
| Male | 2 (40) | 3 (100) | 2 (100) | |
| **Race** | | | | >0.9 |
| Asian | 0 (0) | 0 (0) | 0 (0) | |
| Black or African American | 1 (20) | 1 (50) | 1 (50) | |
| Hispanic or Latino | 0 (0) | 0 (0) | 0 (0) | |
| Other | 0 (0) | 0 (0) | 0 (0) | |
| White | 4 (80) | 1 (50) | 1 (50) | |
| Missing | 0 | 1 | 0 | |
| **Ethnicity** | | | | >0.9 |
| Hispanic/Latino | 1 (25) | 1 (33) | 0 (0) | |
| Non-Hispanic/Latino | 3 (75) | 2 (67) | 2 (100) | |
| Missing | 1 | 0 | 0 | |
| **Gestational age at birth (weeks)** | 27.0 (25.0, 29.4) | 26.6 (26.0, 27.1) | 30.0 (25.0, 35.0) | >0.9 |
| **Birthweight (g)** | 625 (625, 625) | 864 (864, 864) | NA (NA, NA) | 0.3 |
| Missing | 4 | 2 | 2 | |
| **Age at discharge (days)** | 82 (36, 93) | 99 (91, 103) | 93 (66, 121) | 0.6 |
| **Neonatal Hypertension** | 3 (60) | 2 (100) | 1 (50) | >0.9 |
| Missing | 0 | 1 | 0 | |
| **Bronchopulmonary Dysplasia** | 4 (80) | 2 (100) | 1 (50) | >0.9 |
| Missing | 0 | 1 | 0 | |
| **Patent Ductus Arteriosus** | 2 (40) | 0 (0) | 0 (0) | >0.9 |
| Missing | 0 | 1 | 0 | |
| **Congenital heart disease** | 2 (40) | 1 (50) | 1 (50) | >0.9 |
| Missing | 0 | 1 | 0 | |
| **Pulmonary hypertension** | 0 (0) | 0 (0) | 0 (0) | N/A |
| Missing | 0 | 1 | 0 | |

[1] n (%); Median (Q1, Q3).

[2] Fisher's exact test; Kruskal-Wallis rank sum test; NA.

doses higher than the recommended starting dose of 0.01 to 0.05 mg/kg, and two patients exceeded the recommended maximum dose of 0.5 mg/kg every 6–24 hours [15].

Esmolol was used for a shorter duration and titrated more frequently than the oral agents because intravenous agents are easier to titrate to achieve desired blood pressures. Esmolol is likely more often used for more severe hypertension or in younger patients with an inability to tolerate oral medication. In addition, it is probable that patients were initiated on IV therapy acutely prior to transitioning to oral agents for maintenance of blood pressure. The dosing recommendations for esmolol [13] were derived from studies in full-term neonates following cardiac surgery [32]. In that study, esmolol was started at 0.05 mg/kg/min in patients with a postnatal age between 0–7 days and then titrated to achieve a SBP < 90th percentile for age. A more recent study evaluated the use of doses as high as 0.5 mg/kg/min, which was the highest dose seen in our study [33].

Although this study contributes to our knowledge of antihypertensive dosing patterns in preterm infants, there were notable limitations. Because this study did not follow patients longitudinally, it could not evaluate therapeutic outcomes

**Table 5. Summary statistics across patient clusters for dosing patterns of esmolol. Results are presented for the 3-cluster models.**

| Characteristic | A N=9[1] | B N=3[1] | C N=2[1] | p-value[2] |
|---|---|---|---|---|
| **Gender** | | | | 0.7 |
| Female | 7 (78) | 2 (67) | 1 (50) | |
| Male | 2 (22) | 1 (33) | 1 (50) | |
| **Race** | | | | >0.9 |
| Asian | 0 (0) | 0 (0) | 0 (0) | |
| Black or African American | 4 (44) | 1 (50) | 1 (50) | |
| Hispanic or Latino | 0 (0) | 0 (0) | 0 (0) | |
| Other | 0 (0) | 0 (0) | 0 (0) | |
| White | 5 (56) | 1 (50) | 1 (50) | |
| Missing | 0 | 1 | 0 | |
| **Ethnicity** | | | | >0.9 |
| Hispanic/Latino | 0 (0) | 0 (0) | 0 (0) | |
| Non-Hispanic/Latino | 9 (100) | 3 (100) | 1 (100) | |
| Missing | 0 | 0 | 1 | |
| **Gestational age at birth (weeks)** | 34.0 (29.0, 36.4) | 36.0 (31.0, 36.3) | 35.7 (35.4, 36.0) | 0.8 |
| **Birthweight (g)** | 2354 (1600, 2740) | 3747 (3747, 3747) | NA (NA, NA) | 0.14 |
| Missing | 4 | 2 | 2 | |
| **Age at discharge (days)** | 25 (19, 53) | 41 (9, 50) | 56 (10, 101) | >0.9 |
| **Neonatal Hypertension** | 0 (0) | 0 (0) | 0 (0) | N/A |
| **Bronchopulmonary Dysplasia** | 1 (11) | 0 (0) | 0 (0) | >0.9 |
| **Patent Ductus Arteriosus** | 4 (44) | 2 (67) | 2 (100) | 0.6 |
| **Congenital heart disease** | 3 (33) | 2 (67) | 2 (100) | 0.3 |
| **Pulmonary hypertension** | 0 (0) | 0 (0) | 0 (0) | N/A |

[1] n (%); Median (Q1, Q3).

[2] Fisher's exact test; Kruskal-Wallis rank sum test.

of these neonates. Future cohort studies are needed to assess the effectiveness and safety of these different dosing patterns. Our study may lack generalizability to other preterm infants being treated with antihypertensive medications because it was conducted at a single center, and this center generally treats a healthier population of preterm infants (e.g., no infants on extracorporeal membrane oxygenation) than their collaborating NICU at the nearby children's hospital. However, the UAB Medical Center is a regional institution with a Level IV NICU, resulting in a relatively large sample size for a single-center study in this population. In addition, the treatment of hypertension was defined in this study using the proxy of receiving any of 14 antihypertensive agents, possibly resulting in case misclassification because some agents may be used for indications other than hypertension. For example, propranolol is commonly used in infants for hemangiomas. We attempted to mitigate this misclassification by excluding participants with a discharge diagnosis of hemangioma in any code position. Additionally, a prior study of over 100,000 NICU admissions found that receipt of any of these medications had a positive predictive value of 75% for a diagnosis of hypertension [24]. To mitigate these limitations in future studies, a multicenter, prospective study with definitions of hypertension made *a priori* and documentation of the indication the therapy is used for should be conducted. A multicenter study would be representative of the dosing practices used by a larger number of clinicians and NICUs. In addition, a larger sample size will provide greater precision in the identification of true clusters of patients based on antihypertensive dosing trajectories and have higher statistical power to detect any between-group differences among those clusters. The high proportion of patients missing birthweight when querying their electronic

medical records using the UAB clinical data warehouse likely prevented detection of statistically significant differences by birthweight across some clusters.

A strength of this study was its minute-level dosing information. In addition, because there is little knowledge on the dosing trajectories of antihypertensive agents in preterm infants, a strength of this study was its innovative research question and study design. Instead of forming groups *a priori* based on hypothesized dosing strategies, which would exclude patients who did not exhibit those patterns, a data-driven unsupervised machine learning approach was implemented that described all observed dosing patterns. Finally, this study was retrospective, so it required fewer resources and was completed relatively quickly without placing additional ethical burden on patients and their families.

## Conclusions

This study provides an improved understanding of the pharmacotherapeutic strategies routinely used in preterm infants with probable hypertension. It specifically provided additional information regarding the doses of medications being used and the timing for when treatment decisions were made. The first step in determining the optimal dosing of agents is to understand what doses are currently being used. Therefore, the results of this study can be used to determine which agents and dosing strategies would have sufficient sample sizes to be compared in observational comparative effectiveness studies. Further multicenter studies evaluating the dosing strategies used by other providers are needed to identify the optimal therapeutic strategies for preterm infants with probable hypertension.

## Supporting information

**S1 File. File containing one supplemental figure and two supplemental tables.**
(DOCX)

## Acknowledgments

We would like the thank Ashley Severin (UNC Medical Center, Department of Pharmacy) for help identifying antihypertensive drugs to focus on in this analysis. We would like to acknowledge the University of Alabama at Birmingham Medical Center for providing the retrospective data used in this study.

## Author contributions

**Conceptualization:** Mary-Carty Pittman, Alejandro D. Perez, Keia R. Sanderson, Matthew M. Laughon, Matthew Shane Loop.

**Data curation:** Kaniz Afroz Tanni, Jieun Park, Matthew Shane Loop.

**Formal analysis:** Mary-Carty Pittman, Kaniz Afroz Tanni, Jieun Park, Matthew Shane Loop.

**Investigation:** Mary-Carty Pittman, Kaniz Afroz Tanni, Keia R. Sanderson, Jieun Park, Matthew Shane Loop.

**Methodology:** Mary-Carty Pittman, Alejandro D. Perez, Kaniz Afroz Tanni, Keia R. Sanderson, Jieun Park, Matthew Shane Loop.

**Project administration:** Matthew Shane Loop.

**Software:** Mary-Carty Pittman, Kaniz Afroz Tanni, Jieun Park, Matthew Shane Loop.

**Supervision:** Daniel I. Feig, Matthew M. Laughon, Matthew Shane Loop.

**Validation:** Keia R. Sanderson, Daniel I. Feig, Matthew Shane Loop.

**Visualization:** Mary-Carty Pittman, Jieun Park, Matthew Shane Loop.

**Writing – original draft:** Mary-Carty Pittman, Alejandro D. Perez.

**Writing – review & editing:** Mary-Carty Pittman, Alejandro D. Perez, Kaniz Afroz Tanni, Keia R. Sanderson, Jieun Park, Daniel I. Feig, Matthew M. Laughon, Matthew Shane Loop.

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
