## [Decision Letter · Decision Letter 0]

2 Aug 2025

Dear Dr. Loop,

Thank you for submitting your manuscript to PLOS ONE. After careful consideration, we feel that it has merit but does not fully meet PLOS ONE’s publication criteria as it currently stands. Therefore, we invite you to submit a revised version of the manuscript that addresses the points raised during the review process.

We look forward to receiving your revised manuscript.

Kind regards,

Gaetano Santulli, MD, PhD

Academic Editor

PLOS ONE

Journal Requirements:

We would like to acknowledge the University of Alabama at Birmingham Medical Center for providing the retrospective data used in this study. Research reported in this publication was supported by the National Center for Advancing Translational Sciences of the National Institutes of Health under award number UL1TR003096. The content is solely the responsibility of the authors and does not necessarily represent the official views of the National Institutes of Health.

We would like to acknowledge the University of Alabama at Birmingham Medical Center for providing the retrospective data used in this study. Research reported in this publication was supported by the National Center for Advancing Translational Sciences of the National Institutes of Health under award number UL1TR003096. The content is solely the responsibility of the authors and does not necessarily represent the official views of the National Institutes of Health.

We would like to acknowledge the University of Alabama at Birmingham Medical Center for providing the retrospective data used in this study. Research reported in this publication was supported by the National Center for Advancing Translational Sciences of the National Institutes of Health under award number UL1TR003096. The content is solely the responsibility of the authors and does not necessarily represent the official views of the National Institutes of Health.

4. In the online submission form, you indicated that the study authors are prevented from sharing the data publicly due to restrictions placed upon use of the data by the data owners. To initiate a request for the dataset, please contact ResearchData@uabmc.edu and reference request “i2b2.lds.inpatient infants receiving antihypertensives w/dx codes <37 pt set”. The corresponding author (matthew_loop@auburn.edu) will facilitate obtainment of the data for reasonable requests.

5. Please upload a copy of Figure 4, to which you refer in your text on page 11. If the figure is no longer to be included as part of the submission please remove all reference to it within the text.

Reviewers' comments:

Reviewer's Responses to Questions

**Comments to the Author**

1. Is the manuscript technically sound, and do the data support the conclusions?

Reviewer #1: Yes

Reviewer #2: Yes

Reviewer #3: Yes

Reviewer #4: Partly

2. Has the statistical analysis been performed appropriately and rigorously?

Reviewer #1: Yes

Reviewer #2: Yes

Reviewer #3: Yes

Reviewer #4: No

3. Have the authors made all data underlying the findings in their manuscript fully available?

Reviewer #1: Yes

Reviewer #2: Yes

Reviewer #3: Yes

Reviewer #4: Yes

4. Is the manuscript presented in an intelligible fashion and written in standard English?

Reviewer #1: Yes

Reviewer #2: Yes

Reviewer #3: Yes

Reviewer #4: Yes

Reviewer #1: Thank you for the opportunity to review this timely and clinically relevant manuscript titled:

“Dosing trajectories of antihypertensive agents among preterm infants: A retrospective, cross-sectional analysis.”

The study addresses an important gap in neonatal pharmacotherapy by characterizing real-world dosing patterns for antihypertensive agents in a preterm population. The authors appropriately apply functional K-means clustering to time-based dosing data, and the manuscript is generally well-written and well-structured. However, several clarifications and improvements are needed to enhance the interpretability and generalizability of the findings.

Minor Revisions

1. Justification for study population criteria (Page 3, Lines 59–60; Page5, Lines 122-123 )

Please explain why infants with a gestational age <37 weeks and a postmenstrual age <44 weeks were selected. While these thresholds are reasonable, a brief justification would enhance clarity for readers less familiar with NICU conventions.

2. Limitations of cross-sectional design (Page 3, Line 59; Page 5, Line 120; Page 18, Lines 327–335)

As the study is cross-sectional, it does not assess outcomes such as blood pressure control or adverse effects. This limits the ability to draw clinical conclusions about dosing effectiveness. Please acknowledge this more explicitly in the Discussion section.

Suggested addition:

“Because this study does not follow patients longitudinally, it cannot evaluate therapeutic outcomes or causal effects. Future cohort studies are needed to assess the effectiveness and safety of specific dosing patterns.”

3. Small sample sizes in some clusters (Page 13–14)

Captopril and esmolol clusters include very small subgroups (e.g., n=2). Please add a cautionary note that these subgroup findings may not be generalizable.

4. Define "Dosing Trajectory" (Title)

Consider briefly defining “dosing trajectory” early in the Introduction for clarity (e.g., “a patient’s pattern of dose administration over time”). Many readers are not familiar with this terminology.

5. Language and Clarity

Minor grammatical issues are present. Examples:

• Page 4, Line 84: “...hypertension problematic to identify...” → “...making hypertension difficult to identify...”

A brief language review would improve readability. But this is just my suggestions.

6. Figure S1 is not included in the text (pdf file). (Page 8; Line 188) If possible, please provide it to me later.

This is a valuable, data-rich study that provides important descriptive insights into antihypertensive prescribing in neonates. With minor revisions, it will make a strong contribution to neonatal literature.

Reviewer #2: Firstly, thank you for this interesting look at data in a very understudied population. This data involves a degree of conjecture around modeling but it is an absolutely vital first step in getting to drug dosing that better balances safety and efficacy and lays the groundwork for more controlled studies. It can significantly inform day-to-day clinical practice (including my own). The quality of this manuscript overall is very good. Thank you!

Introduction:

I really don’t have a lot to comment on in this section. Overall, I believe it is sufficient in terms of describing the landscape and is well written. Normally I would ask for the authors to speak to existing literature—however, for this particular topic, much of what is available is very old or consensus/editorial rather than robust analysis. I appreciate the insight into publication rates and the issues with data in this population.

The biggest question I had was the choice of agents to analyze—specifically captopril, propranolol, and esmolol. The methods may be a better place for this information but I did not see it discussed there either. Were the top 3 just allocated for k means analysis arbitrarily? Was there a percentage cutoff for which would be included? Is there any description of what other drugs were used and their prevalence?

Was there a conscious choice to exclude diuretics? Indication overlap (most notable with BPD)? Better established dosing? Was usage too small? These are not mentioned at all in the list of drugs cross referenced. While not likely to be the most efficacious drugs their use in clinical practice for mild hypertension in neonates is definitely common, particularly loops and thiazides. I think at least addressing these drug classes and any exclusion rationale is warranted. Amlodipine is another that comes to mind, but that was included in the list of drugs used to identify potential patients to include and perhaps just didn’t have enough usage to be included?

Methods:

125-128: what was the exclusion of IV propranolol for? I see it on the supplemental materials as well and it just sticks out as there weren’t other very specific exclusions like this. Large difference between PO and IV dosing or the short timeframes of administration? Though a broad exclusion, I would concur with the authors on excluding hemangioma—even if it is possible that patients might have coincidentally had propranolol received for hypertension and a hemangioma diagnosis code without pharmacologic treatment of the hemangioma…would be far too confounding to leave in.

133-134: While I get what the gist was, the statement “the primary outcome of this study was the weight-based dose of antihypertensive…” is not technically correct. I think the outcome here is the characterization of doses administered, whether it be trajectory, distribution, etc. If one is calling something an outcome, I think the description of that needs to be technically correct and very narrow, though others may disagree with me here.

135-139: The dose standardization methodology and also addressing weight appears to be appropriate.

140-142: Just to be clear—if no additional doses were given in 6 hours would the data point for dosing trajectory purposes cease there? With as many of these agents that are dosed far less frequently than every 6 hours, I’m trying to understand how this impacts the flow of the data. If a patient is getting a medication every 12 hours, for example, then each dose would be its own point and not be carried forward?

155-165: This explanation of the follow-up time makes sense. Thank you for including.

179-185: This might be an editorial question, but does this information concerning data requests need to be included in the text of the manuscript?

Results:

For all tables, it would be helpful to remove units, including percents, from the numbers reported…for example table 1 might be “ethnicity, n (%)” or as you have done using the footnote to denote units. In that case, they don’t need to be included again in the data reported and it can just be “84 (98)” or similar. The specificity of rounding and sig figs seems to vary—can this be streamlined? Also, some clarification on what is being reported re: formatting. For birthweight in particular, the use of commas in confusing as to what numbers I’m looking at. The IQR is 897 to 2400 or are you reporting 897, 2, and 2400?

“Percent of invasive SBPs above the 90th percentile up to 1 day prior to the first drug administration (%) based on 1987 task force” is so long to include as an item in a table—can this be simplified and the description be noted as a superscript and then a footnote under the table? I also question how much these SBP endpoints actually add to the descriptive analysis here. I can understand why it might be included, but in the context of this study it seems to be a little extraneous, especially given there is a decent amount of missing data for this. Could it be summarized in the text to characterize the clinical appropriateness of therapy without needing to include so much detail?

Table 2: Again, the items in the table are kind of lengthy, can you specify the dosing units for the different drugs in a footnote instead of repeating them in each row?

236-242: Concise. Great! I would avoid using dashes for ranges and use the actual word “to” to clean up numbers in text.

Discussion:

Largely OK.

Line 318 where you start talking about strengths of the study comes off oddly to me, as most studies don’t explain why their work is strong and rather state the strengths in context of the discussion…I don’t if other reviewers would agree, but I would consider reworking that. I do appreciate the detail given to limitations as well!

Reviewer #3: Methods

1- Page 5, Line 128. No clear rationale is provided for excluding IV propranolol if it was used for hypertension, potentially biasing the study by omitting a relevant treatment modality. The authors must explicitly justify this exclusion.

2- Page 6, Line 135-140. Treating IV boluses and infusions identically as mg/kg/minute overlooks distinct pharmacokinetic profiles and clinical strategies, potentially obscuring important nuances in dosing trajectories. The authors must justify this methodological choice

3- Page 7, Lines 164-165. The ad hoc visual inspection was used to choose follow-up cutoffs (1,000 hours for propranolol, 400 hours for captopril, and 250 hours for esmolol). The authors must provide a rationale for these cutoffs

4- The manuscript lacks objective criteria or statistical measures for selecting the optimal number of clusters for each drug

Results

1- Pages 13, 15; Tables 3b, 3c. The small sample sizes in some clusters for captopril (N=2, N=3) and esmolol (N=2, N=3) severely limit the reliability and interpretability of findings for these drugs.

2- Pages 13, 15. The high percentage of Unknown values for critical variables such as birthweight in the cluster analyses is problematic, especially when birthweight is highlighted as a differentiating factor. The authors must discuss the impact of this missing data on validity and generalizability

3- Page 12, line 237. initial doses ff 0.1−0.2 mg/kg. "ff" Should be "of".

Discussion

1- Only 10% of patients had a formal hypertension diagnosis despite all receiving antihypertensives. This significant discrepancy between medication use and formal diagnosis is underexplored.

Reviewer #4: General Assessment

The manuscript addresses a relevant and under-explored topic in the literature. The authors’ work represents a meaningful contribution to the field and has the potential to advance understanding in this area. However, several important methodological and presentation-related issues should be addressed to improve the clarity and rigor of the study.

Clarification of Inclusion Criteria

The manuscript frequently refers to the “treatment of hypertension” in the studied cohort. However, based on the inclusion criteria described, it appears that patients were selected based on receiving antihypertensive treatment, not on a confirmed diagnosis. This distinction is important and should be clarified. Furthermore, Table 1 does not provide sufficient data to verify the presence of hypertension, with only five patient visits meeting the diagnostic definition.

Presentation of Table 1

The current presentation of Table 1 is episode-based, which may obscure important patient-level information. To enhance clarity and provide a more accurate understanding of the cohort, I recommend restructuring Table 1 to present data at the patient level. This would allow for a clearer depiction of comorbidities, particularly when multiple conditions are present in the same individual.

Statistical Support for Group Comparisons

Several statements in the manuscript suggest differences between groups without providing statistical evidence. These assertions should either be supported by appropriate statistical tests or rephrased to avoid implying significance where none has been demonstrated.

Development of Cluster Analysis

The cluster analysis section would benefit from further elaboration. Incorporating ad hoc statistical tests (e.g. v-test) could enhance the interpretability and robustness of the findings, allowing for more meaningful insights to be drawn from the data.

Minor Comments

Consider improving the overall readability of the tables by simplifying formatting and ensuring consistency in terminology (e.g. yes/no/missing data)

Recommendation

I encourage the authors to consider these points in a revised version of the manuscript.

**Do you want your identity to be public for this peer review?** For information about this choice, including consent withdrawal, please see our Privacy Policy

Reviewer #1: No

Reviewer #2: No

Reviewer #3: No

Reviewer #4: No

---

## [Author Response · Author response to Decision Letter 1]

22 Sep 2025

Editorial comments

We would like to acknowledge the University of Alabama at Birmingham Medical Center for providing the retrospective data used in this study. Research reported in this publication was supported by the National Center for Advancing Translational Sciences of the National Institutes of Health under award number UL1TR003096. The content is solely the responsibility of the authors and does not necessarily represent the official views of the National Institutes of Health.

Done

We would like to acknowledge the University of Alabama at Birmingham Medical Center for providing the retrospective data used in this study. Research reported in this publication was supported by the National Center for Advancing Translational Sciences of the National Institutes of Health under award number UL1TR003096. The content is solely the responsibility of the authors and does not necessarily represent the official views of the National Institutes of Health.

We would like to acknowledge the University of Alabama at Birmingham Medical Center for providing the retrospective data used in this study. Research reported in this publication was supported by the National Center for Advancing Translational Sciences of the National Institutes of Health under award number UL1TR003096. The content is solely the responsibility of the authors and does not necessarily represent the official views of the National Institutes of Health.

We have removed the funding statement from the Manuscript and included our updated funding statement in our Cover Letter.

4. In the online submission form, you indicated that the study authors are prevented from sharing the data publicly due to restrictions placed upon use of the data by the data owners. To initiate a request for the dataset, please contact ResearchData@uabmc.edu and reference request “i2b2.lds.inpatient infants receiving antihypertensives w/dx codes <37 pt set”. The corresponding author (matthew_loop@auburn.edu) will facilitate obtainment of the data for reasonable requests.

Thank you. We have included the following text in the appropriate box on the resubmission.

“These data cannot be shared publicly for legal reasons. The data use agreement signed between the authors and the University of Alabama at Birmingham prevents the authors from sharing the data publicly.”

5. Please upload a copy of Figure 4, to which you refer in your text on page 11. If the figure is no longer to be included as part of the submission please remove all reference to it within the text.

We apologize for this oversight. We have removed reference to Figure 4 from the text.

Done

Thank you for your detailed editorial comments.

Reviewer 1

Thank you for the opportunity to review this timely and clinically relevant manuscript titled:

“Dosing trajectories of antihypertensive agents among preterm infants: A retrospective, cross-sectional analysis.”

The study addresses an important gap in neonatal pharmacotherapy by characterizing real-world dosing patterns for antihypertensive agents in a preterm population. The authors appropriately apply functional K-means clustering to time-based dosing data, and the manuscript is generally well-written and well-structured.

We thank the reviewers for their kind and encouraging comments.

However, several clarifications and improvements are needed to enhance the interpretability and generalizability of the findings.

Minor Revisions

1. Justification for study population criteria (Page 3, Lines 59–60; Page5, Lines 122-123 ) Please explain why infants with a gestational age <37 weeks and a postmenstrual age <44 weeks were selected. While these thresholds are reasonable, a brief justification would enhance clarity for readers less familiar with NICU conventions.

We thank the reviewer for recommending we provide a little more detail on why these cutoffs are important for non-NICU familiar readers. We have added text to the beginning of the Methods section, and we reproduce it below for the reviewer’s convenience:

“Our ideal target population was preterm (born < 37 weeks gestation) neonates (< 44 weeks postmenstrual age) treated for hypertension. Postmenstrual age is the age of the infant since the mother’s last menstrual period, and it includes both time spent in the womb (gestational age) plus time outside the womb (i.e., postnatal age). We chose this population because preterm infants have higher risk for hypertension,[21,22] and hypertension often resolves as infant age increases.[23]”

2. Limitations of cross-sectional design (Page 3, Line 59; Page 5, Line 120; Page 18, Lines 327–335) As the study is cross-sectional, it does not assess outcomes such as blood pressure control or adverse effects. This limits the ability to draw clinical conclusions about dosing effectiveness. Please acknowledge this more explicitly in the Discussion section. Suggested addition: “Because this study does not follow patients longitudinally, it cannot evaluate therapeutic outcomes or causal effects. Future cohort studies are needed to assess the effectiveness and safety of specific dosing patterns.”

We thank the reviewer and agree that we should have made this limitation more explicit. We have included the following text at the beginning of the “Limitations” paragraph:

“Because this study did not follow patients longitudinally, it could not evaluate therapeutic outcomes of these neonates. Future cohort studies are needed to assess the effectiveness and safety of these different dosing patterns.”

3. Small sample sizes in some clusters (Page 13–14) Captopril and esmolol clusters include very small subgroups (e.g., n=2). Please add a cautionary note that these subgroup findings may not be generalizable.

We agree and thank the reviewer for this comment. In light of this reviewer’s comment, as well as other reviewers, we have decided to implement null hypothesis significance testing for comparisons among clusters. Therefore, we have removed any statements describing variations across clusters in our sample that did not result in statistically significant inference. Therefore, we believe adding the cautionary note is no longer needed.

4. Define "Dosing Trajectory" (Title) Consider briefly defining “dosing trajectory” early in the Introduction for clarity (e.g., “a patient’s pattern of dose administration over time”). Many readers are not familiar with this terminology.

We thank the reviewer for this wise suggestion. The first sentence of the second paragraph of the Introduction now reads as follows:

“Treating hypertension with a specific pattern of dose administration over time (i.e. a “dose trajectory”) in preterm infants is challenging because there are no treatment guidelines or formal statements from the American Academy of Pediatrics, American College of Cardiology, or American Heart Association.”

5. Language and Clarity Minor grammatical issues are present. Examples:

• Page 4, Line 84: “...hypertension problematic to identify...” → “...making hypertension difficult to identify...” A brief language review would improve readability. But this is just my suggestions.

We thank the reviewer for this comment. We have addressed this specific comment, as well as performed additional language review to improve readability.

6. Figure S1 is not included in the text (pdf file). (Page 8; Line 188) If possible, please provide it to me later.

We apologize that the reviewer was not able to review the Figure S1. We have included it here below for the reviewer’s convenience.

Supplemental Figure 1. Flow Diagram for inclusion and exclusion of patient visits.

This is a valuable, data-rich study that provides important descriptive insights into antihypertensive prescribing in neonates. With minor revisions, it will make a strong contribution to neonatal literature.

We again thank the reviewer for these encouraging comments, and we agree that this manuscript could be a strong contribution to the neonatal literature, especially after including the reviewer’s recommendations.

Reviewer 2

Firstly, thank you for this interesting look at data in a very understudied population. This data involves a degree of conjecture around modeling but it is an absolutely vital first step in getting to drug dosing that better balances safety and efficacy and lays the groundwork for more controlled studies. It can significantly inform day-to-day clinical practice (including my own). The quality of this manuscript overall is very good. Thank you!

We wish to sincerely thank the reviewer for their kind words. We agree that this manuscript involves some conjecture, but that research has to start somewhere. We are glad that the reviewer found the manuscript helpful.

Introduction:

I really don’t have a lot to comment on in this section. Overall, I believe it is sufficient in terms of describing the landscape and is well written. Normally I would ask for the authors to speak to existing literature—however, for this particular topic, much of what is available is very old or consensus/editorial rather than robust analysis. I appreciate the insight into publication rates and the issues with data in this population.

The biggest question I had was the choice of agents to analyze—specifically captopril, propranolol, and esmolol. The methods may be a better place for this information but I did not see it discussed there either. Were the top 3 just allocated for k means analysis arbitrarily? Was there a percentage cutoff for which would be included? Is there any description of what other drugs were used and their prevalence?

We thank the reviewer for their question. We have now included a table of prevalences of all the target drugs observed in our study sample in the supplemental material as Supplemental Table 1. We regret not including this table in the first submission as supplemental material. We include the table below for the reviewer’s convenience:

Characteristic N = 1141

Antihypertensive Drug

Propranolol 70 (61%)

Esmolol 14 (12%)

Captopril 10 (8.8%)

Sodium nitroprusside 6 (5.3%)

Enalapril 5 (4.4%)

Hydralazine 5 (4.4%)

Clonidine 4 (3.5%)

1n (%)

Was there a conscious choice to exclude diuretics? Indication overlap (most notable with BPD)? Better established dosing? Was usage too small? These are not mentioned at all in the list of drugs cross referenced. While not likely to be the most efficacious drugs their use in clinical practice for mild hypertension in neonates is definitely common, particularly loops and thiazides. I think at least addressing these drug classes and any exclusion rationale is warranted. Amlodipine is another that comes to mind, but that was included in the list of drugs used to identify potential patients to include and perhaps just didn’t have enough usage to be included?

We thank the reviewer for their detailed thoughts and comments about the selection of agents to look for. To begin with, we looked for the following agents: vasodilators (hydralazine, nitroprusside, clonidine), angiotensin-converting enzyme (ACE) inhibitors (enalapril, captopril), beta-blockers (propranolol, labetalol, esmolol, atenolol, metoprolol), and calcium channel blockers (amlodipine, isradipine, nifedipine, nicardipine). Any agents that did not appear in the prevalence table above were not identified in our analysis sample (e.g., amlodipine). Follow-up with the University of Alabama at Birmingham confirmed that atenolol and isradipine are non-formulary at that institution for this patient population.

We identified these medications based on the medications identified in a large, multi-site study of antihypertensive drug exposure in NICUs using the Pediatrix database (Ravisankar et al. 2017), as well as additional team input. According to the Ravisankar et al (2017) manuscript, they excluded diuretics “due to their frequent use for other indications (such as, prevention and treatment of bronchopulmonary dysplasia)” (p. 3). As our manuscript was a first attempt to identify dosing trajectories, we decided remain consistent with prior literature on excluding diuretics. Additionally, because our case identification was based on the medication received, it was critical to try and avoid including cases where a particular drug was used for a non-hypertensive indication. Therefore, the exclusion of diuretics helped us to exclude cases where the drug was used for bronchopulmonary dysplasia and not hypertension, at the cost of having a smaller sample size for our study.

To make this exclusion of diuretics more clear for the reader, we have included the follow text in the Methods section, right after the list of drugs we are targeting:

“We did not include diuretics in order to be consistent with prior literature on antihypertensive drug exposure,[24] since including diuretics would have likely identified many cases of bronchopulmonary dysplasia.”

Ravisankar S, Kuehn D, Clark RH, Greenberg RG, Smith PB, Hornik CP. Antihypertensive drug exposure in premature infants from 1997 to 2013. Cardiol Young. 2017;27: 905–911.

Methods:

125-128: what was the exclusion of IV propranolol for? I see it on the supplemental materials as well and it just sticks out as there weren’t other very specific exclusions like this. Large difference between PO and IV dosing or the short timeframes of administration? Though a broad exclusion, I would concur with the authors on excluding hemangioma—even if it is possible that patients might have coincidentally had propranolol received for hypertension and a hemangioma diagnosis code without pharmacologic treatment of the hemangioma…would be far too confounding to leave in.

We thank the reviewer for their comment. IV propranolol was excluded because only 2 patients received propranolol via this route. Because oral propranolol is not highly bioavailable (~25%), oral and IV doses are not equivalent and thus canno

---

## [Decision Letter · Decision Letter 1]

13 Oct 2025

Dear Dr. Loop,

Thank you for submitting your manuscript to PLOS ONE. After careful consideration, we feel that it has merit but does not fully meet PLOS ONE’s publication criteria as it currently stands. Therefore, we invite you to submit a revised version of the manuscript that addresses the points raised during the review process.

We look forward to receiving your revised manuscript.

Kind regards,

Gaetano Santulli, MD

Academic Editor

PLOS ONE

Journal Requirements:

Reviewers' comments:

Reviewer's Responses to Questions

**Comments to the Author**

Reviewer #1: All comments have been addressed

Reviewer #3: All comments have been addressed

Reviewer #4: All comments have been addressed

2. Is the manuscript technically sound, and do the data support the conclusions?

Reviewer #1: Yes

Reviewer #3: Yes

Reviewer #4: Partly

3. Has the statistical analysis been performed appropriately and rigorously?

Reviewer #1: Yes

Reviewer #3: Yes

Reviewer #4: (No Response)

4. Have the authors made all data underlying the findings in their manuscript fully available?

Reviewer #1: Yes

Reviewer #3: Yes

Reviewer #4: Yes

5. Is the manuscript presented in an intelligible fashion and written in standard English?

Reviewer #1: Yes

Reviewer #3: Yes

Reviewer #4: Yes

Reviewer #1: I appreciate the revisions; the authors have fully addressed my comments. I hope this study will serve as a valuable reference for the care of preterm neonates.

Reviewer #3: (No Response)

Reviewer #4: The authors have done a great job of taking the reviewers' comments into account, which is commendable. For my part, a minor revision should be done in the conclusion. The authors should be more cautious with the assertion “with hypertension” and replace it with “with probable hypertension.”

**Do you want your identity to be public for this peer review?** For information about this choice, including consent withdrawal, please see our Privacy Policy

Reviewer #1: **Yes: ** Kwan Young Hong

Reviewer #3: No

Reviewer #4: No

---

## [Author Response · Author response to Decision Letter 2]

17 Oct 2025

Reviewer #1: I appreciate the revisions; the authors have fully addressed my comments. I hope this study will serve as a valuable reference for the care of preterm neonates.

We thank the reviewer for their encouragement, and we are glad that our changes fully addressed the suggestions. The paper is much better now.

Reviewer #4: The authors have done a great job of taking the reviewers' comments into account, which is commendable. For my part, a minor revision should be done in the conclusion. The authors should be more cautious with the assertion “with hypertension” and replace it with “with probable hypertension.”

We thank the reviewer for their encouragement. We have changed the two instances in the conclusion from “hypertension” to “probable hypertension”. The new Conclusions section is reproduced below for the reviewer’s convenience:

“This study provides an improved understanding of the pharmacotherapeutic strategies routinely used in preterm infants with probable hypertension. It specifically provided additional information regarding the doses of medications being used and the timing for when treatment decisions were made. The first step in determining the optimal dosing of agents is to understand what doses are currently being used. Therefore, the results of this study can be used to determine which agents and dosing strategies would have sufficient sample sizes to be compared in observational comparative effectiveness studies. Further multicenter studies evaluating the dosing strategies used by other providers are needed to identify the optimal therapeutic strategies for preterm infants with probable hypertension.”

---

## [Editor Report · Decision Letter 2]

4 Nov 2025

Dosing trajectories of antihypertensive agents among preterm neonates: A retrospective, cross-sectional analysis

PONE-D-25-33596R2

Dear Dr. Loop,

We’re pleased to inform you that your manuscript has been judged scientifically suitable for publication and will be formally accepted for publication once it meets all outstanding technical requirements.

Kind regards,

Gaetano Santulli, MD

Academic Editor

PLOS ONE

---

## [Editor Report · Acceptance letter]

PONE-D-25-33596R2

PLOS ONE

Dear Dr. Loop,

I'm pleased to inform you that your manuscript has been deemed suitable for publication in PLOS ONE. Congratulations! Your manuscript is now being handed over to our production team.

Kind regards,

on behalf of

Professor Gaetano Santulli

Academic Editor

PLOS ONE